# Understanding Probabilistic Sparse Gaussian Process Approximations

**Matthias Bauer**[†‡]        **Mark van der Wilk**[†]        **Carl Edward Rasmussen**[†]

[†]Department of Engineering, University of Cambridge, Cambridge, UK
[‡]Max Planck Institute for Intelligent Systems, Tübingen, Germany
{msb55, mv310, cer54}@cam.ac.uk

## Abstract

Good sparse approximations are essential for practical inference in Gaussian Processes as the computational cost of exact methods is prohibitive for large datasets. The Fully Independent Training Conditional (FITC) and the Variational Free Energy (VFE) approximations are two recent popular methods. Despite superficial similarities, these approximations have surprisingly different theoretical properties and behave differently in practice. We thoroughly investigate the two methods for regression both analytically and through illustrative examples, and draw conclusions to guide practical application.

## 1 Introduction

Gaussian Processes (GPs) [1] are a flexible class of probabilistic models. Perhaps the most prominent practical limitation of GPs is that the computational requirement of an exact implementation scales as $O(N^3)$ time, and as $O(N^2)$ memory, where $N$ is the number of training cases. Fortunately, recent progress has been made in developing *sparse approximations*, which retain the favourable properties of GPs but at a lower computational cost, typically $O(NM^2)$ time and $O(NM)$ memory for some chosen $M < N$. All sparse approximations rely on focussing inference on a small number of quantities, which represent approximately the entire posterior over functions. These quantities can be chosen differently, e.g., function values at certain input locations, properties of the spectral representations [2], or more abstract representations [3]. Similar ideas are used in random feature expansions [4, 5].

Here we focus on methods that represent the approximate posterior using the function value at a set of *M inducing inputs* (also known as pseudo-inputs). These methods include the Deterministic Training Conditional (DTC) [6] and the Fully Independent Training Conditional (FITC) [7], see [8] for a review, as well as the Variational Free Energy (VFE) approximation [9]. The methods differ both in terms of the theoretical approach in deriving the approximation, and in terms of how the inducing inputs are handled. Broadly speaking, inducing inputs can either be chosen from the training set (e.g. at random) or be optimised over. In this paper we consider the latter, as this will generally allow for the best trade-off between accuracy and computational requirements. Training the GP entails jointly optimizing over inducing inputs and hyperparameters.

In this work, we aim to thoroughly investigate and characterise the difference in behaviour of the FITC and VFE approximations. We investigate the biases of the bounds when learning hyperparameters, where each method allocates its modelling capacity, and the optimisation behaviour. In Section 2 we briefly introduce inducing point methods and state the two algorithms using a unifying notation. In Section 3 we discuss properties of the two approaches, both theoretical and practical. Our aim is to understand the approximations in detail in order to know under which conditions each method is likely to succeed or fail in practice. We highlight issues that may arise in practical situations and how to diagnose and possibly avoid them. Some of the properties of the methods have been previously

reported in the literature; our aim here is a more complete and comparative approach. We draw conclusions in Section 4.

## 2   Sparse Gaussian Processes

A Gaussian Process is a flexible distribution over functions, with many useful analytical properties. It is fully determined by its mean $m(\mathbf{x})$ and covariance $k(\mathbf{x}, \mathbf{x}')$ functions. We assume the mean to be zero, without loss of generality. The covariance function determines properties of the functions, like smoothness, amplitude, etc. A finite collection of function values at inputs $\{\mathbf{x}_i\}$ follows a Gaussian distribution $\mathcal{N}(\mathbf{f}; 0, K_{\mathbf{ff}})$, where $[K_{\mathbf{ff}}]_{ij} = k(\mathbf{x}_i, \mathbf{x}_j)$.

Here we revisit the GP model for regression [1]. We model the function of interest $f(\cdot)$ using a GP prior, and noisy observations at the input locations $X = \{\mathbf{x}_i\}_i$ are observed in the vector $\mathbf{y}$.

$$p(\mathbf{f}) = \mathcal{N}(\mathbf{f}; 0, K_{\mathbf{ff}}) \qquad\qquad p(\mathbf{y}|\mathbf{f}) = \prod_{n=1}^{N} \mathcal{N}\left(y_n; f_n, \sigma_n^2\right) \qquad (1)$$

Throughout, we employ a squared exponential covariance function $k(x, x') = s_f^2 \exp(-\frac{1}{2}|x - x'|^2/\ell^2)$, but our results only rely on the decay of covariances with distance. The hyperparameter $\theta$ contains the signal variance $s_f^2$, the lengthscale $\ell$ and the noise variance $\sigma_n^2$, and is suppressed in the notation.

To make predictions, we follow the common approach of first determining $\theta$ by optimising the marginal likelihood and then marginalising over the posterior of $\mathbf{f}$:

$$\theta^* = \underset{\theta}{\arg\max}\, p(\mathbf{y}|\theta) \qquad p(y^*|\mathbf{y}) = \frac{p(y^*, \mathbf{y})}{p(\mathbf{y})} = \int p(y^*|f^*)p(f^*|\mathbf{f})p(\mathbf{f}|\mathbf{y})\mathrm{d}\mathbf{f}\mathrm{d}f^* \qquad (2)$$

While the marginal likelihood, the posterior and the predictive distribution all have closed-form Gaussian expressions, the cost of evaluating them scales as $\mathcal{O}(N^3)$ due to the inversion of $K_{\mathbf{ff}} + \sigma_n^2 I$, which is impractical for many datasets.

Over the years, the two inducing point methods that have remained most influential are FITC [7] and VFE [9]. Unlike previously proposed methods (see [6, 10, 8]), both FITC and VFE provide an approximation to the marginal likelihood which allows both the hyperparameters and inducing inputs to be learned from the data through gradient based optimisation. Both methods rely on the low rank matrix $Q_{\mathbf{ff}} = K_{\mathbf{fu}}K_{\mathbf{uu}}^{-1}K_{\mathbf{uf}}$ instead of the full rank $K_{\mathbf{ff}}$ to reduce the size of any matrix inversion to $M$. Note that for most covariance functions, the eigenvalues of $K_{\mathbf{uu}}$ are not bounded away from zero. Any practical implementation will have to address this to avoid numerical instability. We follow the common practice of adding a tiny diagonal *jitter* term $\varepsilon I$ to $K_{\mathbf{uu}}$ before inverting.

### 2.1   Fully Independent Training Conditional (FITC)

Over the years, FITC has been formulated in several different ways. A form of FITC first appeared in an online learning setting by Csató and Opper [11], derived from the viewpoint of approximating the full GP posterior. Snelson and Ghahramani [7] introduced FITC as approximate inference in a model with a modified likelihood and proposed using its marginal likelihood to train the hyperparameters and inducing inputs jointly. An alternate interpretation where the prior is modified, but exact inference is performed, was presented in [8], unifying it with other techniques. The latest interesting development came with the connection that FITC can be obtained by approximating the GP posterior using Expectation Propagation (EP) [12, 13, 14].

Using the interpretation of modifying the prior to

$$p(\mathbf{f}) = \mathcal{N}(\mathbf{f}; 0, Q_{\mathbf{ff}} + \mathrm{diag}[K_{\mathbf{ff}} - Q_{\mathbf{ff}}]) \qquad (3)$$

we obtain the objective function in Eq. (5). We would like to stress, however, that this modification gives *exactly* the same procedure as approximating the full GP posterior with EP. Regardless of the fact that that FITC *can* be seen as a completely different model, we aim to characterise it as an approximation to the full GP.

## 2.2 Variational Free Energy (VFE)

Variational inference can also be used to approximate the true posterior. We follow the derivation by Titsias [9] and bound the marginal likelihood, by instantiating extra function values on the latent Gaussian process $\mathbf{u}$ at locations $Z$,[1] followed by lower bounding the marginal likelihood. To ensure efficient calculation, $q(\mathbf{u}, \mathbf{f})$ is chosen to factorise as $q(\mathbf{u})p(\mathbf{f}|\mathbf{u})$. This removes terms with $K_{\mathbf{ff}}^{-1}$:

$$\log p(\mathbf{y}) \geq \int q(\mathbf{u}, \mathbf{f}) \log \frac{p(\mathbf{y}|\mathbf{f})\cancel{p(\mathbf{f}|\mathbf{u})}p(\mathbf{u})}{\cancel{p(\mathbf{f}|\mathbf{u})}q(\mathbf{u})} \, d\mathbf{u} \, d\mathbf{f} \qquad (4)$$

The optimal $q(\mathbf{u})$ can be found by variational calculus resulting in the lower bound in Eq. (5).

## 2.3 Common notation

The objective functions for both VFE and FITC look very similar. In the following discussion we will refer to a common notation of their negative log marginal likelihood (NLML) $\mathcal{F}$, which will be minimised to train the methods:

$$\mathcal{F} = \frac{N}{2}\log(2\pi) + \underbrace{\frac{1}{2}\log|Q_{\mathbf{ff}} + G|}_{\text{complexity penalty}} + \underbrace{\frac{1}{2}\mathbf{y}^{\mathsf{T}}(Q_{\mathbf{ff}} + G)^{-1}\mathbf{y}}_{\text{data fit}} + \underbrace{\frac{1}{2\sigma_n^2}\operatorname{tr}(T)}_{\text{trace term}}, \qquad (5)$$

where

$$G_{\text{FITC}} = \operatorname{diag}[K_{\mathbf{ff}} - Q_{\mathbf{ff}}] + \sigma_n^2 I \qquad G_{\text{VFE}} = \sigma_n^2 I \qquad (6)$$
$$T_{\text{FITC}} = 0 \qquad\qquad\qquad T_{\text{VFE}} = K_{\mathbf{ff}} - Q_{\mathbf{ff}}. \qquad (7)$$

The common objective function has three terms, of which the data fit and complexity penalty have direct analogues to the full GP. The **data fit** term penalises the data lying outside the covariance ellipse $Q_{\mathbf{ff}} + G$. The **complexity penalty** is the integral of the data fit term over all possible observations $\mathbf{y}$. It characterises the *volume* of possible datasets that are compatible with the data fit term. This can be seen as the mechanism of *Occam's razor* [16], by penalising the methods for being able to predict too many datasets. The **trace term** in VFE ensures that the objective function is a true lower bound to the marginal likelihood of the full GP. Without this term, VFE is identical to the earlier DTC approximation [6] which can grossly over-estimate the marginal likelihood. The trace term penalises the sum of the conditional variances at the training inputs, conditioned on the inducing inputs [17]. Intuitively, it ensures that VFE not only models this specific dataset $\mathbf{y}$ well, but also approximates the covariance structure of the full GP $K_{\mathbf{ff}}$.

# 3 Comparative behaviour

As our main test case we use the one dimensional dataset[2] considered in [7, 9] with 200 input-output pairs. Of course, sparse methods are not necessary for this toy problem, but all of the issues we raise are illustrated nicely in this one dimensional task which can easily be plotted. In Sections 3.1 to 3.3 we illustrate issues relating to the *objecctive functions*. These properties are independent of how the method is optimised. However, whether they are encountered in practice can depend on optimiser dynamics, which we discuss in Sections 3.4 and 3.5.

## 3.1 FITC can severely underestimate the noise variance, VFE overestimates it

In the full GP with Gaussian likelihood we assume a homoscedastic (input *in*dependent) noise model with noise variance parameter $\sigma_n^2$. It fully characterises the uncertainty left after completely learning the latent function. In this section we show how FITC can also use the diagonal term $\operatorname{diag}(K_{\mathbf{ff}} - Q_{\mathbf{ff}})$ in $G_{\text{FITC}}$ as heteroscedastic (input dependent) noise [7] to account for these differences, thus, invalidating the above interpretation of the noise variance parameter. In fact, the FITC objective function encourages underestimation of the noise variance, whereas the VFE bound encourages overestimation. The latter is in line with previously reported biases of variational methods [18].

Fig. 1 shows the configuration most preferred by the FITC objective for a subset of 100 data points of the Snelson dataset, found by an exhaustive manual search for a minimum over hyperparameters,

inducing inputs *and* number of inducing points. The noise variance is shrunk to practically zero, despite the mean prediction not going through every data point. Note how the mean still behaves well and how the training data lie well within the predictive variance. Only when considering predictive probabilities will this behaviour cause diminished performance. VFE, on the other hand, is able to approximate the posterior predictive distribution almost exactly.

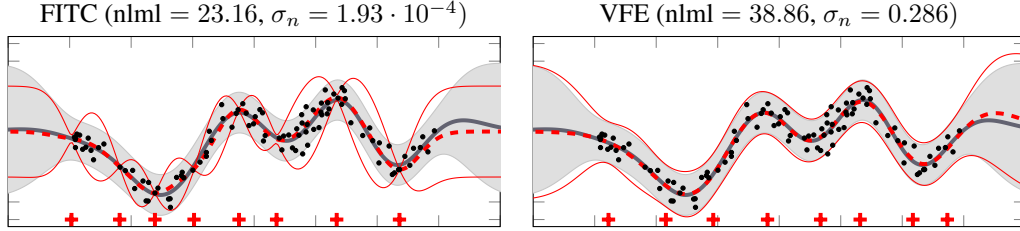

Figure 1: Behaviour of FITC and VFE on a subset of 100 data points of the Snelson dataset for 8 inducing inputs (red crosses indicate inducing inputs; red lines indicate mean and $2\sigma$) compared to the prediction of the full GP in grey. Optimised values for the full GP: nlml $= 34.15$, $\sigma_n = 0.274$

For both approximations, the complexity penalty decreases with decreased noise variance, by reducing the volume of datasets that can be explained. For a full GP and VFE this is accompanied by a data fit penalty for data points lying far away from the predictive mean. FITC, on the other hand, has an additional mechanism to avoid this penalty: its diagonal correction term $\mathrm{diag}(K_{\mathbf{ff}} - Q_{\mathbf{ff}})$. This term can be seen as an input dependent or heteroscedastic noise term (discussed as a modelling advantage by Snelson and Ghahramani [7]), which is zero exactly at an inducing input, and which grows to the prior variance away from an inducing input. By placing the inducing inputs near training data that happen to lie near the mean, the heteroscedastic noise term is locally shrunk, resulting in a reduced complexity penalty. Data points both far from the mean and far from inducing inputs do not incur a data fit penalty, as the heteroscedastic noise term has increased around these points. This mechanism removes the need for the homoscedastic noise to explain deviations from the mean, such that $\sigma_n^2$ can be turned down to reduce the complexity penalty further.

This explains the extreme pinching (severely reduced noise variance) observed in Fig. 1, also see, e.g., [9, Fig. 2]. In examples with more densely packed data, there may not be any places where a near-zero noise point can be placed without incurring a huge data-fit penalty. However, inducing inputs will be placed in places where the data happens to randomly cluster around the mean, which still results in a decreased noise estimate, albeit less extreme, see Figs. 2 and 3 where we use all 200 data points.

**Remark 1** *FITC has an alternative mechanism to explain deviations from the learned function than the likelihood noise and will underestimate $\sigma_n^2$ as a consequence. In extreme cases, $\sigma_n^2$ can incorrectly be estimated to be almost zero.*

As a consequence of this additional mechanism, $\sigma_n^2$ can no longer be interpreted in the same way as for VFE or the full GP. $\sigma_n^2$ is often interpreted as the amount of uncertainty in the dataset which can not be explained. Based on this interpretation, a low $\sigma_n^2$ is often used as an indication that the dataset is being fitted well. Active learning applications rely on a similar interpretation to differentiate between inherent noise, and uncertainty in the latent GP which can be reduced. FITC's different interpretation of $\sigma_n^2$ will cause efforts like these to fail.

VFE, on the other hand, is biased towards over-estimating the noise variance, because of both the data fit and the trace term. $Q_{\mathbf{ff}} + \sigma_n^2 I$ has $N - M$ eigenvectors with an eigenvalue of $\sigma_n^2$, since the rank of $Q_{\mathbf{ff}}$ is $M$. Any component of $\mathbf{y}$ in these directions will result in a larger data fit penalty than for $K_{\mathbf{ff}}$, which can only be reduced by increasing $\sigma_n^2$. The trace term can also be reduced by increasing $\sigma_n^2$.

**Remark 2** *The VFE objective tends to over-estimate the noise variance compared to the full GP.*

### 3.2   VFE improves with additional inducing inputs, FITC may ignore them

Here we investigate the behaviour of each method when more inducing inputs are added. For both methods, adding an extra inducing input gives it an extra basis function to model the data with. We discuss how and why VFE always improves, while FITC may deteriorate.

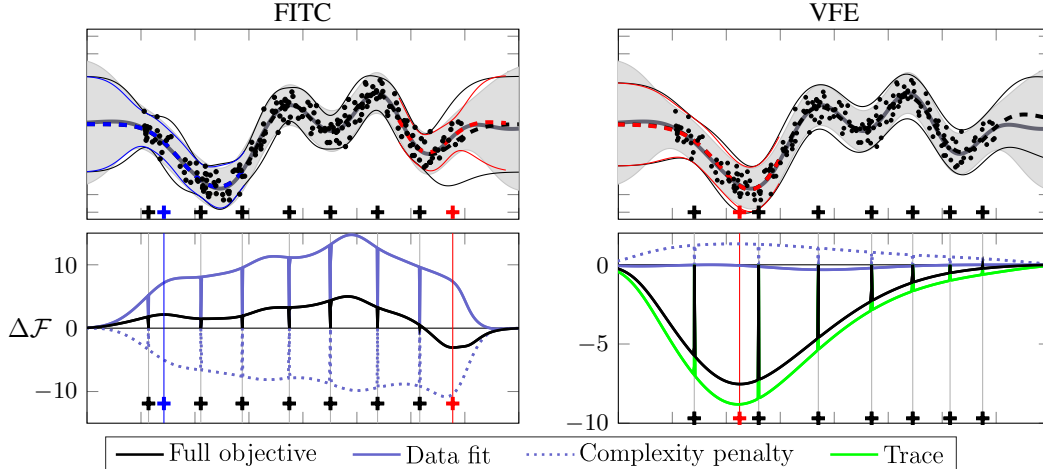

Figure 2: *Top:* Fits for FITC and VFE on 200 data points of the Snelson dataset for $M = 7$ optimised inducing inputs (black). *Bottom:* Change in objective function from adding an inducing input anywhere along the $x$-axis (no further hyperparameter optimisation performed). The overall change is decomposed into the change in the individual terms (see legend). Two particular additional inducing inputs and their effect on the predictive distribution shown in red and blue.

Fig. 2 shows an example of how the objective function changes when an inducing input is added anywhere in the input domain. While the change in objective function looks reasonably smooth overall, there are pronounced spikes for both, FITC and VFE. These return the objective to the value without the additional inducing input and occur at the locations of existing inducing inputs. We discuss the general change first before explaining the spikes.

Mathematically, adding an inducing input corresponds to a rank 1 update of $Q_{\mathbf{ff}}$, and can be shown to always improve VFE's bound[3], see Supplement for a proof. VFE's complexity penalty increases due to an extra non-zero eigenvalue in $Q_{\mathbf{ff}}$, but gains in data fit and trace.

**Remark 3** *VFE's posterior and marginal likelihood approximation become more accurate (or remain unchanged) regardless of where a new inducing input is placed.*

For FITC, the objective can change either way. Regardless of the change in objective, the heteroscedastic noise is decreased at all points (see Supplement for proof). For a squared exponential kernel, the decrease is strongest around the newly placed inducing input. This decrease has two effects. One, it reduces the complexity penalty since the diagonal component of $Q_{\mathbf{ff}} + G$ is reduced and replaced by a more strongly correlated $Q_{\mathbf{ff}}$. Two, it worsens the data fit term as the heteroscedastic term is required to fit the data when the homoscedastic noise is underestimated. Fig. 2 shows reduced error bars with several data points now outside of the 95% prediction bars. Also shown is a case where an additional inducing input improves the objective, where the extra correlations outweigh the reduced heteroscedastic noise.

Both VFE and FITC exhibit pathological behaviour (spikes) when inducing inputs are clumped, that is, when they are placed exactly on top of each other. In this case, the objective function has the same value as when all duplicate inducing inputs were removed, see Supplement for a proof. In other words, for all practical purposes, a model with duplicate inducing inputs reduces to a model with fewer, individually placed inducing inputs.

Theoretically, these pathologies only occur at single points, such that no gradients towards or away from them could exist and they would never be encountered. In practise, however, these peaks are widend by a finite *jitter* that is added to $K_{\mathbf{uu}}$ to ensure it remains well conditioned enough to be invertible. This finite width provides the gradients that allow an optimiser to detect these configurations.

As VFE always improves with additional inducing inputs, these configurations must correspond to maxima of the optimisation surface and clumping of inducing inputs does not occur for VFE. For

FITC, configurations with clumped inducing inputs can and often do correspond to minima of the optimisation surface. By placing them on top of each other, FITC can avoid the penalty of adding an extra inducing input and can gain the bonus from the heteroscedastic noise. Clumping, thus, constitutes a mechanism that allows FITC to effectively remove inducing inputs at no cost.

We illustrate this behaviour in Fig. 3 for 15 randomly initialised inducing inputs. FITC places some of them exactly on top of each other, whereas VFE spreads them out and recovers the full GP well.

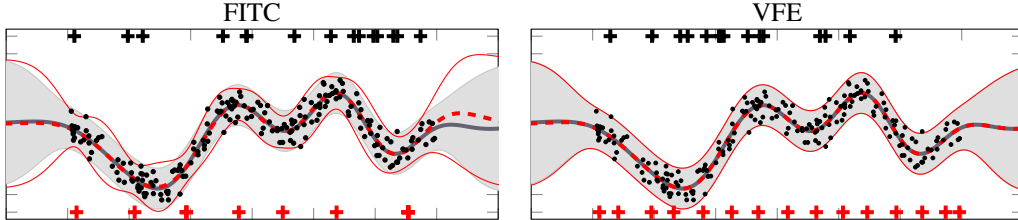

Figure 3: Fits for 15 inducing inputs for FITC and VFE (initial as black crosses, optimised red crosses). Even following joint optimisation of inducing inputs and hyperparameters, FITC avoids the penalty of added inducing inputs by clumping some of them on top of each other (shown as a single red cross). VFE spreads out the inducing inputs to get closer to the true full GP posterior.

**Remark 4** *In FITC, having a good approximation $Q_{\mathbf{ff}}$ to $K_{\mathbf{ff}}$ needs to be traded off with the gains coming from the heteroscedastic noise. FITC does not always favour a more accurate approximation to the GP.*

**Remark 5** *FITC avoids losing the gains of the heteroscedastic noise by placing inducing inputs on top of each other, effectively removing them.*

### 3.3 FITC does not recover the full GP posterior, VFE does

In the previous section we showed that FITC may not utilise additional resources to model the data. The clumping behaviour, thus, explains why the FITC objective may not recover the full GP, even when given enough resources.

Both VFE and FITC *can* recover the true posterior by placing an inducing input on every training input [9, 12]. For VFE, this is a *global* minimum, since the KL gap to the true marginal likelihood is zero. For FITC, however, this configuration is not stable and the objective can still be improved by clumping of inducing inputs, as Matthews [19] has shown empirically by aggressive optimisation. The derivative of the inducing inputs is zero for the initial configuration, but adding jitter subtly makes this behaviour more obvious by perturbing the gradients, similar to the widening of the peaks in Fig. 2. In Fig. 4 we reproduce the observations in [19, Sec 4.6.1 and Fig. 4.2] on a subset of 100 data points of the Snelson dataset: VFE remains at the minimum and, thus, recovers the full GP, whereas FITC improves its objective and clumps the inducing inputs considerably.

| Method | nml initial | nml optimised |
|--------|-------------|---------------|
| Full GP | – | 33.8923 |
| VFE | 33.8923 | 33.8923 |
| FITC | 33.8923 | 28.3869 |

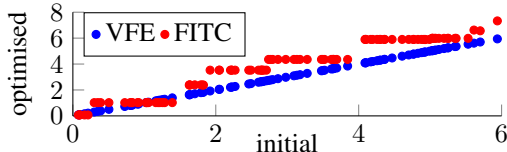

Figure 4: Results of optimising VFE and FITC after initialising at the solution that gives the correct posterior and marginal likelihood as in [19, Sec 4.6.1]: FITC moves to a significantly different solution with better objective value (Table, *left*) and clumped inducing inputs (Figure, *right*).

**Remark 6** *FITC generally does not recover the full GP, even when it has enough resources.*

### 3.4 FITC relies on local optima

So far, we have observed some cases where FITC fails to produce results in line with the full GP, and characterised why. However, in practice, FITC has performed well, and pathological behaviour is not always observed. In this section we discuss the optimiser dynamics and show that they help FITC behave reasonably.

To demonstrate this behaviour, we consider a 4d toy dataset: 1024 training and 1024 test samples drawn from a 4d Gaussian Process with isotropic squared exponential covariance function ($l = 1.5, s_f = 1$) and true noise variance $\sigma_n^2 = 0.01$. The data inputs were drawn from a Gaussian centred around the origin, but similar results were obtained for uniformly sampled inputs. We fit both FITC and VFE to this dataset with the number of inducing inputs ranging from 16 to 1024, and compare a representative run to the full GP in Fig. 5.

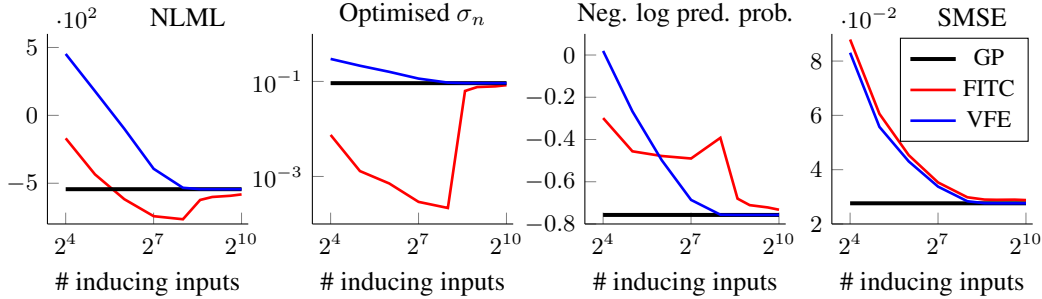

Figure 5: Optimisation behaviour of VFE and FITC for varying number of inducing inputs compared to the full GP. We show the objective function (negative log marginal likelihood), the optimised noise $\sigma_n$, the negative log predictive probability and standardised mean squared error as defined in [1].

VFE monotonically approaches the values of the full GP but initially overestimates the noise variance, as discussed in Section 3.1. Conversely, we can identify three regimes for the objective function of FITC: 1) Monotonic improvement for few inducing inputs, 2) a region where FITC over-estimates the marginal likelihood, and 3) recovery towards the full GP for many inducing inputs. Predictive performance follows a similar trend, first improving, then declining while the bound is estimated to be too high, followed by a recovery. The recovery is counter to the usual intuition that over-fitting worsens when adding more parameters.

We explain the behaviour in these three regimes as follows: When the number of inducing inputs are severely limited (regime 1), FITC needs to place them such that $K_{\mathbf{ff}}$ is well approximated. This correlates most points to some degree, and ensures a reasonable data fit term. The marginal likelihood is under-estimated due to lack of a flexibility in $Q_{\mathbf{ff}}$. This behaviour is consistent with the intuition that limiting model capacity prevents overfitting.

As the number of inducing inputs increases (regime 2), the marginal likelihood is over-estimated and the noise drastically under-estimated. Additionally, performance in terms of log predictive probability deteriorates. This is the regime closest to FITC's behaviour in Fig. 1. There are enough inducing inputs such that they can be placed such that a bonus can be gained from the heteroscedastic noise, without gaining a complexity penalty from losing long scale correlations.

Finally, in regime 3, FITC starts to behave more like a regular GP in terms of marginal likelihood, predictive performance and noise variance parameter $\sigma_n$. FITC's ability to use heteroscedastic noise is reduced as the approximate covariance matrix $Q_{\mathbf{ff}}$ is closer to the true covariance matrix $K_{\mathbf{ff}}$ when many (initial) inducing input are spread over the input space.

In the previous section we showed that after adding a new inducing input, a better minimum obtained without the extra inducing input could be recovered by clumping. So it is clear that the minimum that was found with fewer active inducing inputs still exists in the optimisation surface of many inducing inputs; the optimiser just does not find it.

**Remark 7** *When running FITC with many inducing inputs its resemblance to the full GP solution relies on local optima, rather than the objective function changing.*

## 3.5 VFE is hindered by local optima

So far we have seen that the VFE objective function is a true lower bound on the marginal likelihood and does not share the same pathologies as FITC. Thus, when optimising, we really are interested in finding a global optimum. The VFE objective function is not completely trivial to optimise, and often tricks, such as initialising the inducing inputs with k-means and initially fixing the hyperparameters

[20, 21], are required to find a good optimum. Others have commented that VFE has the tendency to underfit [3]. Here we investigate the underfitting claim and relate it to optimisation behaviour.

As this behaviour is not observable in our 1D dataset, we illustrate it on the pumadyn32nm dataset[4] (32 dimensions, 7168 training, 1024 test), see Table 1 for the results of a representative run with random initial conditions and $M = 40$ inducing inputs.

| Method | NLML$/N$ | $\sigma_n$ | inv. lengthscales | RMSE |
|---|---|---|---|---|
| GP (SoD) | $-0.099$ | 0.196 | ▮▁▁▁▁▁▁▁▁▁ ⋯ | 0.209 |
| FITC | $-0.145$ | 0.004 | ▮▮▁▁▁▁▁▁▁▁ ⋯ | 0.212 |
| VFE | 1.419 | 1 | ▮▁▮▁▁▁▁▁▁▁ ⋯ | 0.979 |
| VFE (frozen) | 0.151 | 0.278 | ▮▁▁▁▁▁▁▁▁▁ ⋯ | 0.276 |
| VFE (init FITC) | $-0.096$ | 0.213 | ▮▁▁▁▁▁▁▁▁▁ ⋯ | 0.212 |

Table 1: Results for pumadyn32nm dataset. We show negative log marginal likelihood (NLML) divided by number of training points, the optimised noise variance $\sigma_n^2$, the ten most dominant inverse lengthscales and the RMSE on test data. Methods are full GP on 2048 training samples, FITC, VFE, VFE with initially frozen hyperparameters, VFE initialised with the solution obtained by FITC.

Using a squared exponential ARD kernel with separate lengthscales for every dimension, a full GP on a subset of data identified four lengthscales as important to model the data while scaling the other 28 lengthscales to large values (in Table 1 we plot the inverse lengthscales).

FITC was consistently able to identify the same four lengthscales and performed similarly compared to the full GP but scaled down the noise variance $\sigma_n^2$ to almost zero. The latter is consistent with our earlier observations of strong pinching in a regime with low-density data as is the case here due to the high dimensionality. VFE, on the other hand, was unable to identify these relevant lengthscales when jointly optimising the hyperparameters and inducing inputs, and only identified some of the them when initially freezing the hyperparameters. One might say that VFE "underfits" in this case. However, we can show that VFE still *recognises* a good solution: When we initialised VFE with the FITC solution it consistently obtained a good fit to the model with correctly identified lengthscales and a noise variance that was close to the full GP.

**Remark 8** *VFE has a tendency to find under-fitting solutions. However, this is an optimisation issue. The bound correctly identifies good solutions.*

## 4 Conclusion

In this work, we have thoroughly investigated and characterised the differences between FITC and VFE, both in terms of their objective function and their behaviour observed during practical optimisation. We highlight several instances of undesirable behaviour in the FITC objective: over-estimation of the marginal likelihood, sometimes severe under-estimation of the noise variance parameter, wasting of modelling resources and not recovering the true posterior. The common practice of using the noise variance parameter as a diagnostic for good model fitting is unreliable. In contrast, VFE is a true bound to the marginal likelihood of the full GP and behaves predictably: It correctly identifies good solutions, always improves with extra resources and recovers the true posterior when possible. In practice however, the pathologies of the FITC objective do not always show up, thanks to "good" local optima and (unintentional) early stopping. While VFE's objective recognises a good configuration, it is often more susceptible to local optima and harder to optimise than FITC.

Which of these pathologies show up in practise depends on the dataset in question. However, based on the superior properties of the VFE objective function, we recommend using VFE, while paying attention to optimisation difficulties. These can be mitigated by careful initialisation, random restarts, other optimisation tricks and comparison to the FITC solution to guide VFE optimisation.

**Acknowledgements**
We would like to thank Alexander Matthews, Thang Bui, and Richard Turner for useful discussions.

## Footnotes

[1]Matthews et al. [15] show that this procedure approximates the posterior over the entire process $f$ correctly.

[2]Obtained from `http://www.gatsby.ucl.ac.uk/~snelson/`

[3]Matthews [19] independently proved this result by considering the KL divergence between processes. Titsias [9] proved this result for the special case when the new inducing input is selected from the training data.

[4]obtained from `http://www.cs.toronto.edu/~delve/data/datasets.html`

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
