[Supplementary Material]

# Supplemental Material for
# Understanding Probabilistic Sparse
# Gaussian Process Approximations

**Matthias Bauer**[†‡]          **Mark van der Wilk**[†]          **Carl Edward Rasmussen**[†]
[†]Department of Engineering, University of Cambridge, Cambridge, UK
[‡]Max Planck Institute for Intelligent Systems, Tübingen, Germany
{msb55, mv310, cer54}@cam.ac.uk

## 1   Proofs for additional inducing inputs

In this section we give proofs to the claims on how the objective functions for FITC and VFE change upon adding an inducing input. For this, we first restate the objective function:

$$\mathcal{F} = \frac{N}{2}\log(2\pi) + \underbrace{\frac{1}{2}\log|Q_{\mathbf{ff}} + G|}_{\text{complexity penalty}} + \underbrace{\frac{1}{2}\mathbf{y}^{\intercal}(Q_{\mathbf{ff}} + G)^{-1}\mathbf{y}}_{\text{data fit}} + \underbrace{\frac{1}{2\sigma_n^2}\operatorname{tr}(T)}_{\text{trace term}}, \tag{1}$$

where

$$G_{\text{FITC}} = \operatorname{diag}[K_{\mathbf{ff}} - Q_{\mathbf{ff}}] + \sigma_n^2 I \qquad G_{\text{VFE}} = \sigma_n^2 I \tag{2}$$

$$T_{\text{FITC}} = 0 \qquad\qquad\qquad T_{\text{VFE}} = K_{\mathbf{ff}} - Q_{\mathbf{ff}}. \tag{3}$$

$K_{\mathbf{ff}}$ denotes the covariance matrix and $Q_{\mathbf{ff}} = K_{\mathbf{fu}}K_{\mathbf{uu}}^{-1}K_{\mathbf{uf}}$ the approximate covariance matrix. Note that the approximate covariance matrix $Q_{\mathbf{ff}}$ is the only quantity depending on the inducing inputs.

The main results we show in this supplement are:

1. Adding an inducing input corresponds to a rank 1 update of the approximate covariance matrix $Q_{\mathbf{ff}} = K_{\mathbf{fu}}K_{\mathbf{uu}}^{-1}K_{\mathbf{uf}}$ (see Section 1.1)

2. For VFE the objective function can never get worse by adding an inducing input anywhere in the input domain (see Section 1.2)

3. For FITC the heteroscedastic noise always decreases when adding an inducing input anywhere in the input domain (see Section 1.3)

4. Adding an inducing input on top of an existing inducing input does not change the objective function, neither for FITC nor for VFE (see Section 1.4)

### 1.1   Adding an inducing input anywhere

In this section we show that adding an inducing input corresponds to a rank 1 update of the $Q_{\mathbf{ff}}$ matrix for the case without jitter.
Let $K_{\mathbf{uu}}$ denote the covariance matrix of the $M$ inducing inputs. We then add a new $M + 1$st inducing input and denote all quantities depending on the new set of $M + 1$ inducing inputs by a superscript $+$.
The updated approximate covariance matrix $Q_{\mathbf{ff}}^+$ is then given by:

$$Q_{\mathbf{ff}}^+ = K_{\mathbf{fu}}^+ (K_{\mathbf{uu}}^+)^{-1} K_{\mathbf{uf}}^+ \tag{4}$$

We proceed by first computing an explicit expression for the $M+1 \times M+1$ matrix $(K_{\mathbf{uu}}^+)^{-1}$ before computing $Q_{\mathbf{ff}}^+$. For this we employ the block matrix inversion formula[1], see Eq. (8)

$$
\begin{aligned}
(K_{\mathbf{uu}}^+)^{-1} &= \begin{pmatrix} K_{\mathbf{uu}} & k_{\mathbf{u}} \\ k_{\mathbf{u}}^\mathsf{T} & k \end{pmatrix}^{-1} \\
&= \begin{pmatrix} K_{\mathbf{uu}}^{-1} + \frac{1}{c}\mathbf{aa}^\mathsf{T} & -\frac{1}{c}\mathbf{a} \\ -\frac{1}{c}\mathbf{a}^\mathsf{T} & \frac{1}{c} \end{pmatrix}
\end{aligned}
\qquad\qquad \mathbf{a} = K_{\mathbf{uu}}^{-1} k_{\mathbf{u}}
$$

where $c = k - k_{\mathbf{u}}^\mathsf{T} K_{\mathbf{uu}}^{-1} k_{\mathbf{u}}$ is the Schur complement of $K_{\mathbf{uu}}$, $k_{\mathbf{u}}^\mathsf{T} = (k(Z_1, Z_{M+1}), \ldots, k(Z_M, Z_{M+1}))$ is the vector of covariances between the old inducing inputs and the added inducing input and $k = k(Z_{M+1}, Z_{M+1})$ is the covariance function evaluated at the new inducing input. Note that $c$ needs to be non-zero for this expression to make sense.

$$
K_{\mathbf{uf}}^+ = \begin{pmatrix} K_{\mathbf{uf}} \\ k_{\mathbf{f}}^\mathsf{T} \end{pmatrix}
$$

where $k_{\mathbf{f}}^\mathsf{T} = (k(Z_{M+1}, x_1), \ldots, k(Z_{M+1}, x_N))$ is the vector of covariances between the data points and the new inducing input.

We can now compute $Q_{\mathbf{ff}}^+$ by the product in Eq. (4) to see that is is indeed given by a rank 1 update of $Q_{\mathbf{ff}}$

$$
\begin{aligned}
Q_{\mathbf{ff}}^+ &= K_{\mathbf{fu}}^+ (K_{\mathbf{uu}}^+)^{-1} K_{\mathbf{uf}}^+ \\
&= K_{\mathbf{fu}} K_{\mathbf{uu}}^{-1} K_{\mathbf{uf}} + \frac{1}{c} \big( K_{\mathbf{fu}} \mathbf{aa}^\mathsf{T} K_{\mathbf{uf}} + K_{\mathbf{fu}} \mathbf{aa}^\mathsf{T} K_{\mathbf{uf}} \\
&\qquad\qquad\quad -K_{\mathbf{fu}} \mathbf{a} k_{\mathbf{f}}^\mathsf{T} - k_{\mathbf{f}} \mathbf{a}^\mathsf{T} K_{\mathbf{uf}} + k_{\mathbf{f}} k_{\mathbf{f}}^\mathsf{T} \big) \\
&= K_{\mathbf{fu}} K_{\mathbf{uu}}^{-1} K_{\mathbf{uf}} + \frac{1}{c} \left( K_{\mathbf{fu}} \mathbf{a} - k_{\mathbf{f}} \right) \left( K_{\mathbf{fu}} \mathbf{a} - k_{\mathbf{f}} \right)^\mathsf{T} \\
&= K_{\mathbf{fu}} K_{\mathbf{uu}}^{-1} K_{\mathbf{uf}} + \mathbf{bb}^\mathsf{T} \\
&= Q_{\mathbf{ff}} + \mathbf{bb}^\mathsf{T} \\
Q_{\mathbf{ff}}^+ &= Q_{\mathbf{ff}} + \mathbf{bb}^\mathsf{T}
\end{aligned}
\tag{5}
$$

where we have introduced the rank 1 update vector $\mathbf{b} = \frac{1}{\sqrt{c}} (K_{\mathbf{fu}} K_{\mathbf{uu}}^{-1} k_{\mathbf{u}} - k_{\mathbf{f}})$.

Thus, in the case of no *jitter*, the update is indeed given by a rank 1 update. These results also extend to the case with finite jitter, which is then absorbed into the definition of $K_{\mathbf{uu}}$ and $k$, respectively.

## 1.2 The VFE objective function always improves when adding an additional inducing input

We now compute the change in objective function when adding the $M + 1$ st inducing input:

$$
\begin{aligned}
2(\mathcal{F}^+ - \mathcal{F}) &= \log \left| Q_{\mathbf{ff}}^+ + \sigma_n^2 I \right| - \log \left| Q_{\mathbf{ff}} + \sigma_n^2 I \right| + \mathbf{y}^\mathsf{T} (Q_{\mathbf{ff}}^+ + \sigma_n^2 I)^{-1} \mathbf{y} - \mathbf{y}^\mathsf{T} (Q_{\mathbf{ff}} + \sigma_n^2 I)^{-1} \mathbf{y} \\
&\quad + \frac{1}{\sigma_n^2} \operatorname{tr}(K_{\mathbf{ff}} - Q_{\mathbf{ff}}^+) - \frac{1}{\sigma_n^2} \operatorname{tr}(K_{\mathbf{ff}} - Q_{\mathbf{ff}}) \\
&= \log \left| Q_{\mathbf{ff}} + \mathbf{bb}^\mathsf{T} + \sigma_n^2 I \right| - \log \left| Q_{\mathbf{ff}} + \sigma_n^2 I \right| \\
&\quad + \mathbf{y}^\mathsf{T} (Q_{\mathbf{ff}} + \mathbf{bb}^\mathsf{T} + \sigma_n^2 I)^{-1} \mathbf{y} - \mathbf{y}^\mathsf{T} (Q_{\mathbf{ff}} + \sigma_n^2 I)^{-1} \mathbf{y} - \frac{1}{\sigma_n^2} \operatorname{tr}(\mathbf{bb}^\mathsf{T})
\end{aligned}
$$

To deal with the log-determinant-terms and the inverses, we employ the Matrix determinant lemma[2] and the Sherman–Morrison formula[3], respectively

$$2(\mathcal{F}^+ - \mathcal{F}) = \log(1 + \mathbf{b}^\mathsf{T}(Q_{\mathbf{ff}} + \sigma_n^2 I)^{-1}\mathbf{b}) + \log\left|Q_{\mathbf{ff}} + \sigma_n^2 I\right| - \log\left|Q_{\mathbf{ff}} + \sigma_n^2 I\right|$$

$$+ \mathbf{y}^\mathsf{T}(Q_{\mathbf{ff}} + \sigma_n^2 I)^{-1}\mathbf{y} - \mathbf{y}^\mathsf{T}\frac{(Q_{\mathbf{ff}} + \sigma_n^2 I)^{-1}\mathbf{b}\mathbf{b}^T(Q_{\mathbf{ff}} + \sigma_n^2 I)^{-1}}{1 + \mathbf{b}^T(Q_{\mathbf{ff}} + \sigma_n^2 I)^{-1}\mathbf{b}}\mathbf{y}$$

$$- \mathbf{y}^\mathsf{T}(Q_{\mathbf{ff}} + \sigma_n^2 I)^{-1}\mathbf{y} + \frac{1}{\sigma_n^2}\operatorname{tr}(\mathbf{b}\mathbf{b}^\mathsf{T})$$

$$= \log(1 + \mathbf{b}^\mathsf{T}(Q_{\mathbf{ff}} + \sigma_n^2 I)^{-1}\mathbf{b}) - \frac{1}{\sigma_n^2}\operatorname{tr}(\mathbf{b}\mathbf{b}^\mathsf{T})$$

$$- \mathbf{y}^\mathsf{T}\frac{(Q_{\mathbf{ff}} + \sigma_n^2 I)^{-1}\mathbf{b}\mathbf{b}^T(Q_{\mathbf{ff}} + \sigma_n^2 I)^{-1}}{1 + \mathbf{b}^T(Q_{\mathbf{ff}} + \sigma_n^2 I)^{-1}\mathbf{b}}\mathbf{y}$$

We can bound the first two terms by noting

$$\operatorname{tr}(\mathbf{b}\mathbf{b}^\mathsf{T}) = \mathbf{b}^\mathsf{T}\mathbf{b}$$

$$\log(1 + x) \leq x$$

$$\mathbf{b}^\mathsf{T}(Q_{\mathbf{ff}} + \sigma_n^2 I)^{-1}\mathbf{b} \leq \frac{1}{\sigma_n^2}\mathbf{b}^\mathsf{T}\mathbf{b}$$

Thus,

$$\log(1 + \mathbf{b}^\mathsf{T}(Q_{\mathbf{ff}} + \sigma_n^2 I)^{-1}\mathbf{b}) - \frac{1}{\sigma_n^2}\operatorname{tr}(\mathbf{b}\mathbf{b}^\mathsf{T}) \leq 0$$

and equality holds for $\mathbf{b} = 0$, as is the case when both inducing inputs lie on top of each other. It remains to show that the term including the $\mathbf{y}$s (including its sign) is non-positive. This can be shown quite easily:

$$-\mathbf{y}^\mathsf{T}\frac{(Q_{\mathbf{ff}} + \sigma_n^2 I)^{-1}\mathbf{b}\mathbf{b}^T(Q_{\mathbf{ff}} + \sigma_n^2 I)^{-1}}{1 + \mathbf{b}^T(Q_{\mathbf{ff}} + \sigma_n^2 I)^{-1}\mathbf{b}}\mathbf{y} = -\frac{(\mathbf{y}^\mathsf{T}(Q_{\mathbf{ff}} + \sigma_n^2 I)^{-1}\mathbf{b})^2}{1 + \mathbf{b}^T(Q_{\mathbf{ff}} + \sigma_n^2 I)^{-1}\mathbf{b}}$$

$$\leq -(\mathbf{y}^\mathsf{T}(Q_{\mathbf{ff}} + \sigma_n^2 I)^{-1}\mathbf{b})^2$$

$$\leq 0$$

where the second to last inequality holds as $(Q_{\mathbf{ff}} + \sigma_n^2 I)^{-1}$ is positive definite. Equalities hold, again, if $\mathbf{b} = 0$ which corresponds to duplication of an existing inducing input.

This concludes the proof that the VFE objective function always improves or stays the same.

The change of the objective function for FITC is less clear than for VFE. We can give no proofs about the changes in general. In experiments, we observe that the change in the data fit term can be positive or negative, whereas the complexity penalty term seems to always improve by adding an inducing input. We hypothesise that this is indeed the case but cannot give a proof for this claim.

In this section we have assumed that all matrix inverses exist and that for duplication of an inducing input we find $\mathbf{b} = 0$. However, for a duplicate inducing input, the matrix $K_{\mathbf{uu}}$ becomes singular, such that care has to be taken when reasoning. In Section 1.4 we show that these arguments can, indeed, be made rigorous and that $\mathbf{b} = 0$ for duplicate inducing inputs. Thus, when duplicating an inducing input, the VFE and FITC objective functions do not change. For VFE, this configuration corresponds to a maximum of the objective function.

## 1.3 The heteroscedastic noise is decresed when new inducing inputs are added

While the objective function can change either way, the heteroscedastic noise, which is given by $\operatorname{diag}(K_{\mathbf{ff}} - Q_{\mathbf{ff}})$ always decreases or remains the same when a new inducing input is added:

$$\operatorname{diag}(K_{\mathbf{ff}} - Q_{\mathbf{ff}}^+) = \operatorname{diag}(K_{\mathbf{ff}} - (Q_{\mathbf{ff}} + \mathbf{b}\mathbf{b}^\mathsf{T})) \tag{6}$$

$$= \operatorname{diag}(K_{\mathbf{ff}} - Q_{\mathbf{ff}}) - \operatorname{diag}(\mathbf{b}\mathbf{b}^\mathsf{T}) \tag{7}$$

The diagonal elements of $\mathbf{b}\mathbf{b}^\mathsf{T}$ are given by $b_m^2$, which are always larger or equal to zero, such that the heteroscedastic noise always decreases (or stays the same).

## 1.4 Adding an inducing input on top of another inducing input

In this section we show that duplication of an inducing input, that is, placing an additional inducing input on top of an existing one, does not change the approximate covariance matrix: $Q_{\mathbf{ff}}^+ = Q_{\mathbf{ff}}$.

One might be tempted to use Eq. (5) to evaluate the update when placing an additional inducing input on top of an existing one. In that case $K_{\mathbf{uu}}^{-1} k_{\mathbf{u}} = \widehat{e}_M$, where $\widehat{e}_M$ denotes the indicator vector with a one at the $M$th position and zeros otherwise, and $K_{\mathbf{fu}} K_{\mathbf{uu}}^{-1} k_{\mathbf{u}} = k_{\mathbf{f}}$ suggesting $\mathbf{b} = 0$. However, in this case, the Schur complement vanishes as well, $c = 0$.

In the following we show that this reasoning can be made exact by considering a finite *jitter* term $\epsilon I$ added onto $K_{\mathbf{uu}}^+$ before inversion. The result can be expanded to second order in $\epsilon$ and the limit $\epsilon \to 0$ then leads to the desired result. The intuition behind the fact that Eq. (4) is well behaved, even if $K_{\mathbf{uu}}^+$ is singular, is, that the eigenvector of $K_{\mathbf{uu}}^+$ that corresponds to the zero eigenvalue is never excited by the matrix $K_{\mathbf{uf}}^+$ which has a duplicate row. The eigenvector only has two non-zero elements, which have the same absolute value but different signs, thus cancelling with the duplicate rows in $K_{\mathbf{uf}}^+$.

Moreover, we obtain a correction term that scales with the jitter. For reasons of numerical stability, one has to employ some form of regularisation of the (possibly) singular matrix $K_{\mathbf{uu}}$ in practise. One common way that is implemented in many toolboxes is the constant jitter $\epsilon I$ introduced above. We assume that the original $K_{\mathbf{uu}}$ is non-singular for now.

Similarly to before, we again employ the block matrix inversion formula[4] of the following form:

$$\begin{pmatrix} A & B \\ C & D \end{pmatrix}^{-1} = \begin{pmatrix} A^{-1} + A^{-1} B F^{-1} C A^{-1} & -A^{-1} B F^{-1} \\ -F^{-1} C A^{-1} & F^{-1} \end{pmatrix} \tag{8}$$

where $F = D - C A^{-1} B$ is the Schur complement of $A$.

$$(K_{\mathbf{uu}}^+ + \epsilon I_{M+1 \times M+1})^{-1} = \begin{pmatrix} K_{\mathbf{uu}} + \epsilon I_{M \times M} & k_{\mathbf{u}} \\ k_{\mathbf{u}}^{\mathsf{T}} & k + \epsilon \end{pmatrix}^{-1} \tag{9}$$

where $k_{\mathbf{u}}^{\mathsf{T}} = (k(Z_1, Z_M), k(Z_2, Z_M), \dots, k(Z_M, Z_M))$ and $k = k(Z_M, Z_M)$ similarly to before. In order to perform the inversion, we expand the inverses in Eq. (8) to second order in $\epsilon$:

$$(K_{\mathbf{uu}} + \epsilon I)^{-1} \approx K_{\mathbf{uu}}^{-1}(1 - \epsilon K_{\mathbf{uu}}^{-1} + \epsilon^2 K_{\mathbf{uu}}^{-2} - \epsilon^3 K_{\mathbf{uu}}^{-3}) \qquad A^{-1}$$

$$(K_{\mathbf{uu}} + \epsilon I)^{-1} k_{\mathbf{u}} \approx \widehat{e}_M - \epsilon k_{\mathbf{u}}^{-1} + \epsilon^2 K_{\mathbf{uu}}^{-1} k_{\mathbf{u}}^{-1} - \epsilon^3 K_{\mathbf{uu}}^{-2} k_{\mathbf{u}}^{-1} \qquad A^{-1}B$$

$$k_{\mathbf{u}}^T (K_{\mathbf{uu}} + \epsilon I)^{-1} \approx \widehat{e}_M^T - \epsilon (k_{\mathbf{u}}^{-1})^T + \epsilon^2 (k_{\mathbf{u}}^{-1})^T K_{\mathbf{uu}}^{-1} - \epsilon^3 (k_{\mathbf{u}}^{-1})^T K_{\mathbf{uu}}^{-2} \qquad CA^{-1}$$

$$k_{\mathbf{u}}^T (K_{\mathbf{uu}} + \epsilon)^{-1} k_{\mathbf{u}} \approx k - \epsilon + \epsilon^2 k^{-1} - \epsilon^3 (k_{\mathbf{u}}^{-1})^T k_{\mathbf{u}}^{-1} \qquad CA^{-1}B$$

$$k + \epsilon - k_{\mathbf{u}}^T (K_{\mathbf{uu}} + \epsilon)^{-1} k_{\mathbf{u}} \approx 2\epsilon(1 - \frac{\epsilon}{2} k^{-1} + \frac{\epsilon^2}{2} (k_{\mathbf{u}}^{-1})^T k_{\mathbf{u}}^{-1}) \qquad \text{Schur}$$

$$(k + \epsilon - k_{\mathbf{u}}^T (K_{\mathbf{uu}} + \epsilon)^{-1} k_{\mathbf{u}})^{-1} \approx \frac{1}{2\epsilon} + \frac{1}{4} k^{-1} - \frac{\epsilon}{4} (k_{\mathbf{u}}^{-1})^T k_{\mathbf{u}}^{-1} + \frac{\epsilon}{8} (k^{-1})^2 \qquad \text{Schur}^{-1}$$

where $\widehat{e}_M = (0, \dots, 0, 1)^{\mathsf{T}}$ is the indicator vector with a one at position $M$ and $k_{\mathbf{u}}^{-1} = (k^{-1}(Z_1, Z_M), \dots, k^{-1}(Z_M, Z_M))^{\mathsf{T}}$ is the $M$th column of $K_{\mathbf{uu}}^{-1}$. Note that the elements of $k_{\mathbf{u}}^{-1}$ are not element wise inverses but elements of an inverse matrix! Analogously, $k^{-1}$ denotes the $(M, M)$ element of the matrix $K_{\mathbf{uu}}^{-1}$.

$$(K_{\mathbf{uu}}^+ + \epsilon)^{-1} = \left(\begin{array}{ccc|c} & K_{\mathbf{uu}}^{-1} & & \begin{matrix} 0 \\ \vdots \\ 0 \end{matrix} \\ \hline 0 & \cdots & 0 & 0 \end{array}\right) - \epsilon \left(\begin{array}{ccc|c} & K_{\mathbf{uu}}^{-2} & & \begin{matrix} 0 \\ \vdots \\ 0 \end{matrix} \\ \hline 0 & \cdots & 0 & 0 \end{array}\right) + \frac{\epsilon}{2} \left(\begin{array}{ccc|c} & k_{\mathbf{u}}^{-1}(k_{\mathbf{u}}^{-1})^\mathsf{T} & & \begin{matrix} 0 \\ \vdots \\ 0 \end{matrix} \\ \hline 0 & \cdots & 0 & 0 \end{array}\right)$$

$$+ \frac{1}{2} \left(\begin{array}{cccc|c} & & & 0 & 0 \\ & 0_{M-1\times M-1} & & \vdots & \vdots \\ & & & 0 & 0 \\ 0 & \cdots & 0 & \epsilon^{-1} & -\epsilon^{-1} \\ \hline 0 & \cdots & 0 & -\epsilon^{-1} & \epsilon^{-1} \end{array}\right)$$

$$+ \frac{1}{4} \left(\begin{array}{cccc|c} & & & 0 & 0 \\ & 0_{M-1\times M-1} & & \vdots & \vdots \\ & & & 0 & 0 \\ 0 & \cdots & 0 & k^{-1} & -k^{-1} \\ \hline 0 & \cdots & 0 & -k^{-1} & k^{-1} \end{array}\right) + \frac{1}{2} \left(\begin{array}{cc|c} 0_{M-1\times M-1} & -k_{u\backslash m}^{-1} & k_{\mathbf{u}}^{-1} \\ -(k_{u\backslash m}^{-1})^T & -2k^{-1} & \\ \hline (k_{\mathbf{u}}^{-1})^T & & 0 \end{array}\right)$$

$$+ \frac{\epsilon}{4} \left(\begin{array}{cc|c} 0_{M-1\times M-1} & 2K_{\mathbf{uu}}^{-1}k_{\mathbf{u}}^{-1} - k^{-1}k_{\mathbf{u}}^{-1} & -(2K_{\mathbf{uu}}^{-1}k_{\mathbf{u}}^{-1} - k^{-1}k_{\mathbf{u}}^{-1}) \\ (2K_{\mathbf{uu}}^{-1}k_{\mathbf{u}}^{-1} - k^{-1}k_{\mathbf{u}}^{-1})^T & & \\ -(2K_{\mathbf{uu}}^{-1}k_{\mathbf{u}}^{-1} - k^{-1}k_{\mathbf{u}}^{-1})^T & & 0 \end{array}\right)$$

$$+ \frac{\epsilon}{8} \left(\begin{array}{cccc|c} & & & 0 & 0 \\ & 0_{M-1\times M-1} & & \vdots & \vdots \\ & & & 0 & 0 \\ 0 & \cdots & 0 & (k^{-1})^2 - 2(k_{\mathbf{u}}^{-1})^Tk_{\mathbf{u}}^{-1} & -((k^{-1})^2 - 2(k_{\mathbf{u}}^{-1})^Tk_{\mathbf{u}}^{-1}) \\ \hline 0 & \cdots & 0 & -((k^{-1})^2 - 2(k_{\mathbf{u}}^{-1})^Tk_{\mathbf{u}}^{-1}) & (k^{-1})^2 - 2(k_{\mathbf{u}}^{-1})^Tk_{\mathbf{u}}^{-1} \end{array}\right) + \mathcal{O}(\epsilon^2)$$

For the matrix $(K_{\mathbf{uu}} + \epsilon I)^{-1}$ we find:

$$(K_{\mathbf{uu}} + \epsilon)^{-1} = K_{\mathbf{uu}}^{-1} - \epsilon K_{\mathbf{uu}}^{-2} + \mathcal{O}(\epsilon^2)$$

When we now multiply out the product $K_{\mathbf{fu}}^+(K_{\mathbf{uu}}^+ + \epsilon I)^{-1}K_{\mathbf{uf}}^+$, we note that $K_{\mathbf{uf}}^+$ will have a duplicate row and $K_{\mathbf{fu}}^+$ will have a duplicate column. Due to this, all terms that have the submatrix $0_{M-1\times M-1}$ in their upper left hand corner cancel. This includes the term that contains the exploding (in the limit $\epsilon \to 0$) inverse jitter $\epsilon^{-1}$, and we are left with:

$$Q_{\mathbf{ff}}^+ = K_{\mathbf{fu}}^+(K_{\mathbf{uu}}^+ + \epsilon I)^{-1}K_{\mathbf{uf}}^+ = K_{\mathbf{fu}}K_{\mathbf{uu}}^{-1}K_{\mathbf{uf}} - \epsilon K_{\mathbf{fu}}K_{\mathbf{uu}}^{-2}K_{\mathbf{uf}} + \frac{\epsilon}{2}K_{\mathbf{fu}}k_{\mathbf{u}}^{-1}(k_{\mathbf{u}}^{-1})^\mathsf{T}K_{\mathbf{uf}} + \mathcal{O}(\epsilon^2)$$

$$= Q_{\mathbf{ff}} - \epsilon K_{\mathbf{fu}}K_{\mathbf{uu}}^{-2}K_{\mathbf{uf}} + \frac{\epsilon}{2}K_{\mathbf{fu}}k_{\mathbf{u}}^{-1}(k_{\mathbf{u}}^{-1})^\mathsf{T}K_{\mathbf{uf}} + \mathcal{O}(\epsilon^2)$$

Such that the correction to the original approximate covariance matrix is given by:

$$Q_{\mathbf{ff}}^+ - Q_{\mathbf{ff}} = \frac{\epsilon}{2}K_{\mathbf{fu}}k_{\mathbf{u}}^{-1}(k_{\mathbf{u}}^{-1})^\mathsf{T}K_{\mathbf{uf}} + \mathcal{O}(\epsilon^2) \tag{10}$$

We can now take the limit $\epsilon \to 0$ as all the "infinities" have cancelled above. For finite jitter, the correction term is again given by a rank-1 update to first order in $\epsilon$.

## Footnotes

[1] https://en.wikipedia.org/wiki/Block_matrix#Block_matrix_inversion

[2]https://en.wikipedia.org/wiki/Matrix_determinant_lemma

[3]https://en.wikipedia.org/wiki/Sherman-Morrison_formula

[4]https://en.wikipedia.org/wiki/Block_matrix#Block_matrix_inversion