[Reviews · NeurIPS 2016]

Reviewer 1

Summary

This paper provides a careful and detailed analysis of the two most popular approaches for sparse Gaussian process approximations. These are the full independent training conditional (FITC) approximation and Titsias' trick based on a variational free energy (VFE). The paper highlights some of the pathologies that occur when using these two approximations. The paper includes several experiments on toy datasets to illustrate its claims. From the paper one can see that FITC tends to underestimate the predictive variance while VFE does not. On the contrary the authors show that VFE is a more difficult to optimize criterion and that some tricks are needed in some situations to get good results. The authors also show that FITC often benefits from sub-optimal convergence points.

Qualitative Assessment

The paper is very well written and it can be followed very easily. This paper is an important step towards better understanding current state-of-the-art approaches for sparse Gaussian processes. While it is true that many of the results shown by the authors were already known and are present in the literature, this paper provides a really nice summary of them, and besides, it includes new results that were not known previously. I think that the paper could be improved by commenting on the behavior of the different methods in the very large regime. In particular, recent work has shown that one can use stochastic methods to optimize both the FITC and VFE criterion. See [18] and for example: D. Hernandez-Lobato, J. M. Hernandez-Lobato. Scalable Gaussian Process Classification via Expectation Propagation, AISTATS, 2016. Although this paper considers classification. Addressing regression problems should be straight-forward, since the likelihood is already Gaussian in that case. See also: T. Bui, J. Yan, R. E. Turner. A Unifying Framework for Sparse Gaussian Process Approximation using Power Expectation Propagation. http://arxiv.org/pdf/1605.07066.pdf I am wondering if these pathologies will also appear there. Specially, when m << n and n is in the order of millions.

Confidence in this Review

2-Confident (read it all; understood it all reasonably well)


Reviewer 2

Summary

The paper reviews and examines the properties of the FITC and VFE sparse GP approximations. The paper provides valuable insight on the approximations and collects (and in some cases extends) many previously shown results into one place. In summary, the paper is well written and presents results worth publishing.

Qualitative Assessment

I don't have any major comments on the paper. I liked it and I think the results, discussion and conclusions provide valuable information for the community. Many of the issues are such that those who have worked with these approximations extensively are aware of them but here they are summarized in one place Few minor comments. Line 113: what is the prior for the number of inducing inputs. Figure 2. Do you keep length-scale constant when adding the inducing input here, or do you optimize the hyperparameters as well? I assume the former but you could state it clearly. Table 1. Explain how "rmse distance Z" was calculated. Figure 4. - Did you do this comparison only once? You could provide some frequency characteristics to ensure that this does not happen only for this particular data. - How were inputs (x) spanned with respect to the length-scale? The "density" of inputs with respect to the length-scale affects the behavior of the approximations. Related to this, what would the behavior be in higher/lower dimensions if the density of inputs was kept constant. - show the three regimes you mention in the text also in this figure.

Confidence in this Review

3-Expert (read the paper in detail, know the area, quite certain of my opinion)


Reviewer 3

Summary

The paper gives theoretical analysis of FITC and FVE approximations for sparse large-scale Gaussian processes. There are various analytical results and proofs that are interesting.

Qualitative Assessment

The paper analyses the differences between the two main large-scale approximations strategies for GP inference, namely the variational inference of Titsias'09, and the FITC by Snelson'06. It's an interesting paper that tries to summarise how these two main approaches differ and how they affect the resulting models. The paper is limited in that it only covers how these two approximation methods affect the resulting models, but does not comment on the running times at all. The only motivation for sparse approximations is large-scale problems, and thus scalability. Perhaps for this reason they do not consider e.g. EP and SVI approaches, which have their own interesting approximation effects the resulting model as well. A longer format might fit this paper better, with analyses of all currently used approaches instead of just these two. All the analyses are interesting, especially regarding the local minima, heteroscedastic noises and addition of inducing points. 3.4. illustrates beautifully the behaviors of FITC/FVE. It seems that FVE is superior, but then in 3.5. a real dataset shows FITC matching or surpassing VFE performance. To explain this discrepancy more large-scale datasets should have been analysed to see if FITC consistently beats VFE, and in which settings. The theoretical drawbacks of FITC would be cast in strange light if it does not seem to suffer from them in practise, as hinted in table 2. I'm also having problems with the running example regarding FITC. Fig1 shows the FITC to give a badly "pinched" model. The same dataset was used in many other papers (with roughly comparable experimental settings), where this does not happen for FITC. In both Snelson'06 and Titsias'09 the FITC gives a good non-pinched model for this data, and in Quinonero'05 the FITC gives slighly pinched model, but not as bad as in the present paper. This is likely due to number of datapoints (fig3 again is not pinched), so it would be nice to see how the "pinching" effects behaves as a function of data size. Overall it's an excellent paper with good findings on FITC/FVE, but the theory and practise seem to disagree in large-scale cases, and there is no running time analysis. The paper would also be greatly strenghtened by being more comprehensive in compared methods (eg. EP/SVI).

Confidence in this Review

2-Confident (read it all; understood it all reasonably well)


Reviewer 4

Summary

This paper performs a thorough analysis and comparison on two of the most influential approximations of Gaussian processes, both analytically and through illustrative examples. Conclusions, which favor one of the methods, are drawn as a guide to practical applications.

Qualitative Assessment

This paper broadens our understanding on two of the most influential approximations of Gaussian processes, namely, FITC and VFE. A few of the notable facts/points made in the paper include: 1. FITC may severely underestimate the noise. 2. VFE improves the marginal likelihood approximation with additional inducing inputs, whereas FITC may ignore them (because the inducing inputs may overlap and the overlaps do not contribute to the likelihood). 3. VFE progressively better approximates the full GP posterior with more inducing inputs, whereas FITC may overshoot the marginal likelihood. 4. FITC's behavior exhibits three regimes as the number of inducing inputs increases. The paper is well written and in my opinion, fits NIPS publication. In what follows are technical discussions, which should not be considered a negative opinion toward the acceptance of the paper. Whereas the authors indicate a strong theoretical preference of VFE, to be fair, the practical picture heavily relies on the optimization procedure used for estimation. Optimization is hard. Just like what is shown in Section 3.5, local minima may render a superior result for FITC and inferior for VFE. Even when local minima is not an issue in Figure 4, both methods yield very similar prediction error across the whole range of inducing inputs. Moreover, it is arguable whether overshooting the likelihood, by using that of the full GP as a standard, is an undesirable effect in practice.

Confidence in this Review

2-Confident (read it all; understood it all reasonably well)


Reviewer 5

Summary

This paper focuses on two similar methods, i.e., the fully independent training conditional (FITC) and the variational free energy (VFE) approximations. These methods are two widely used method for sparse approximation and can solve the high computational cost problem of Gaussian Process. The major contribution of this paper is a deep analysis of the differences of both FITC and VFE in terms of theoretical properties and practical behavior.

Qualitative Assessment

This paper focuses on two similar methods, i.e., the fully independent training conditional (FITC) and the variational free energy (VFE) approximations. These methods are two widely used method for sparse approximation and can solve the high computational cost problem of Gaussian Process. The major contribution of this paper is a deep analysis of the differences of both FITC and VFE in terms of theoretical properties and practical behavior. Overall, I think it is nice to compare FITC and VFE and identify the differences in the theoretical properties and practical behavior, so that the researchers can choose which one is suitable for specific applications and/or build more efficient and effective methods based on the findings of this paper. These findings include 1) FITC can severely underestimate the noise variance, and VFE overestimates it; 2) VFE improves with additional inducing inputs, and FITC may ignore them; 3) FITC does not recover the true posterior, VFE does; 4) FITC relies on local minima and 5) VFE is hindered by local optima. The analyses are thoroughly and completely. The authors demonstrate the important properties of both methods and compare their performances on different datasets. However, I think it would be better for the authors to point out some promising directions for future research. The paper is well-organized and well-written. However, this paper is not a technically contributed paper. Also, as I am not an expert in this field, it is difficult for me to judge the significance of such comparison, hence, my opinion is borderline.

Confidence in this Review

1-Less confident (might not have understood significant parts)


Reviewer 6

Summary

The paper compares two well known sparse approximations of Gaussian Processes, the Fully Independent Training Conditional (FITC) and the Variational Free Energy (VFE) approximation. Both approximations rely on the idea of using a small number of “inducing inputs” to approximate the posterior. The authors examine both theoretical and practical aspects that separate the two, and give recommendations for which approximation to use. They identify that FITC can severely underestimate noise variance and that VFE overestimates it. They also find that VFE, in contrast to FITC, behaves more nicely when increasing the number of inducing inputs, and that it recovers the true posterior, but that it is hindered by local optima. In conclusion, they recommend using VFE.

Qualitative Assessment

The paper is well structured and well written. It was a pleasure to read. The content is very easy to follow, even for a non-expert, and I think the results are useful for guiding the choice of approximation. The motivation and claims are clear, as well as the support of these claims. Throughout the paper you use the squared exponential covariance function/kernel. I think this is understandable as it is very widely used. To what extent are the results generalizable to other kernels? For example, is the pinching behavior of the FITC aggravated with a Laplacian kernel? Is there are kernel for which it is alleviated? The main dataset used for evaluation is pleasingly simple and small. You discuss several aspects of the results of each method applied to this data and connect these to theoretical aspects of the model, e.g. the pinching of the FITC noise variance. Still, when reading, I cannot help but wonder how general these results are. To some extent, you’ve answered this theoretically, but these results are somewhat hidden. Maybe some of the claims proven in the supplement should be stated in the main paper, and not just as remarks. While you recommend using VFE based on the results of this work, it also has its problems, e.g. optimization difficulties and overestimation of noise variance. Can you, based on your analysis, imagine a novel approximation that overcomes some of the problems of VFE and FITC? If not, do the results suggest the non-existance of such an approximation? There are a few spelling mistakes (e.g. “easilly”. Run spell-check).

Confidence in this Review

2-Confident (read it all; understood it all reasonably well)